# Spontaneous Myocarditis in Mice Predisposed to Autoimmune Disease: Including Vaccination-Induced Onset

**DOI:** 10.3390/biomedicines10061443

**Published:** 2022-06-18

**Authors:** Takuma Hayashi, Motoki Ichikawa, Ikuo Konishi

**Affiliations:** 1School of Medicine, Shinshu University, Nagano 390-8621, Japan; nobuoyaegashi@yahoo.co.jp; 2START-Program, Japan Science and Technology Agency (JST), Tokyo 102-8666, Japan; 3National Hospital Organization Kyoto Medical Centre, Kyoto 612-8555, Japan; ikuokonishi08@yahoo.co.jp; 4Department of Obstetrics and Gynecology, Kyoto University School of Medicine, Kyoto 606-8501, Japan

**Keywords:** myocarditis, NF-κB1, vaccine, nonobese diabetic (NOD), autoimmune disease

## Abstract

Nonobese diabetic (NOD)/ShiLtJ mice, such as biobreeding rats, are used as an animal model for type 1 diabetes. Diabetes develops in NOD mice as a result of insulitis, a leukocytic infiltrate of the pancreatic islets. The onset of diabetes is associated with moderate glycosuria and nonfasting hyperglycemia. Previously, in NOD/ShiLtJ mice spontaneously developing type 1 diabetes, the possible involvement of decreased expression of nuclear factor-kappa B1 (NF-κB1) (also known as p50) in the development of type 1 diabetes was investigated. In response to these arguments, NOD mice with inconsistent NF-κB1 expression were established. Surprisingly, the majority of NOD *Nfκb1* homozygote mice were found to die by the eighth week of life because of severe myocarditis. The incidence of spontaneous myocarditis in mice was slightly higher in males than in females. Furthermore, insulitis was observed in all NOD *Nfκb1* heterozygote mice as early as 4 months of age. Additionally, in NOD *Nfκb1* heterozygote mice, myocarditis with an increase in cTnT levels due to influenza or hepatitis B virus vaccination was observed with no significant gender difference. However, myocarditis was not observed with the two types of human papillomavirus vaccination. The results of immunological assays and histopathological examinations indicated that vaccination could induce myocarditis in genetically modified mice. In this study, we report that NOD *Nfκb1* heterozygote mice can be used for investigating the risk of myocarditis development after vaccination.

## 1. Introduction

Vaccination is anticipated to inhibit viral infection and prevent the progression of infectious diseases. However, for most vaccines, the development of serious disorders, such as anaphylactic shock, a serious allergic disease, and myocarditis, an autoimmune disease, after vaccination is rare [1]. Despite tremendous advances in medical technology, the definitive pathogenic mechanisms and risk factors for serious adverse events after vaccination remain unknown. Recently, severe neurological disorders, such as Guillain–Barré syndrome, have rarely developed after vaccination with human papillomavirus (HPV) vaccines [2]. However, serious side effects observed after HPV vaccination have become a social problem.

During the coronavirus disease 2019 (COVID-19) period, several vaccinations against severe acute respiratory syndrome coronavirus 2 (SARS-CoV-2) have provided significant public health benefits. However, they also have the potential hazards of serious side effects. Recent clinical studies have revealed that the risk of myocarditis increased in multiple age groups of both men and women after vaccination with mRNA-based COVID-19 vaccines (BNT162b2 or mRNA-1273), particularly after the second dose in men aged 12–24 years [3]. However, the risk factors for myocarditis or endocarditis development following mRNA-based COVID-19 vaccines remain unidentified. Vaccination was believed to raise the risk of autoimmune diseases, but no medical evidence has been reported. Therefore, the risks and outcomes of myocarditis after vaccinations with various vaccines, including mRNA-based COVID-19 vaccination, are unclear.

In normal and pathological conditions, the nuclear factor-kappa B (NF-κB) family is an important transcription factor in cells, expressing cytokines, chemokines, growth factors, cell adhesion molecules, and some acute-phase proteins [4]. The NF-κB complex is formed when NF-κB1 or NF-κB2 binds to Rel proto-oncogene (REL), RELA, or RELB. Transgenic mice lacking the *Nfκb1* (also known as p50) subunit of NF-κB did not acquire developmental abnormalities but did exhibit nonspecific responses to infection, such as autoimmunity [5]. Additionally, further studies have revealed that NF-κB1 may be involved in the development of inflammation or autoimmune disease [6,7]. Furthermore, clinical studies indicated that the *-94 del ATTG* polymorphism, which results in *NF-κB1* expression defects, may have functional implications and may be an important risk factor for ulcerative colitis, an immune-mediated, complex genetic disorder [8,9]. On the contrary, novel mutations in *NF-κB1*, *phosphatidylinositol 3-kinase (**PI3Kδ)*, *PI3KR1*, and *protein kinase (**PKCδ)*, which result in the clinical picture of common variable immune deficiency, are highly associated with autoimmune diseases [10,11].

Nonobese diabetic (NOD)/ShiLtJ mice with inconsistent NF-κB1 expression were established. Surprisingly, the majority of NOD *Nfκb1* homozygote mice were found to die by the eighth week of life because of severe myocarditis. Myocarditis occurred in 95% of males and 75% of females, with gender differences. Furthermore, the onset of insulitis was observed in all NOD *Nfκb1* heterozygote mice at 4 months of age. However, the incidence of myocarditis in NOD *Nfκb1* heterozygote mice was low in both males and females (15% for males and 5% for females). Our results indicate that the expression status of NF-κB1 is involved in the onset of diabetes in NOD mice. Furthermore, in F15-NOD *Nfκb1* heterozygote mice, vaccination with influenza HA or hepatitis B virus (HBV) vaccine resulted in an increase in serum cardiac troponin T (cTnT), which is a risk factor for myocarditis, with no gender difference. The F15-NOD *Nfκb1* heterozygote and homozygote mice, which are considered at risk of developing myocarditis, were found to have an increased risk of myocarditis after influenza hemagglutinin (HA) or HBV vaccination. However, no myocarditis was observed with the two types of HPV vaccination. Because of the limited data on the efficacy and safety of all clinically prescribed vaccines, including mRNA-based COVID-19 vaccines, in patients with autoimmune diseases, such as collagen and rheumatic diseases, all clinically prescribed vaccines for patients with autoimmune diseases or people with allergic predisposition should be carefully considered.

## 2. Materials and Methods

### 2.1. Animals

The D2.129P2 (B6)-*Nfκb1* homozygote mice were purchased from the JAX^®^ Mice and Services at the Jackson Laboratory (Bar Harbor, ME, USA). NOD/ShiLtJ mice were purchased from CLEA Japan, Inc. (Meguro, Tokyo, Japan). Additionally, the F15 NOD *Nfκb1* heterozygote mice were created by backcrossing D2.129P2 (B6)-*Nfκb1* homozygote male mice with NOD/ShiLtJ female mice or D2.129P2 (B6)-*Nfκb1* homozygote female mice with NOD/ShiLtJ male mice (F15 means backcrossed more than 15 times). Furthermore, male and female F15 NOD *Nfκb1* heterozygote mice were crossed to create F15 NOD *Nfκb1* homozygote mice.

### 2.2. M-Mode Echocardiography Examination

To determine the disease severity of myocarditis, M-mode echocardiography was performed on mice by SonoSite M (FUJIFILM SonoSite, Inc., Minato-ku, Tokyo, Japan) using standard procedure. Briefly, mice were anesthetized with ether and examined via M-mode echocardiography.

### 2.3. Flow Cytometry

One million splenocytes from D2.129P2 (B6)-*Nfκb1* homozygote mice, NOD/ShiLtJ mice, and F15 NOD *Nfκb1* heterozygote mice were suspended in biotin-free Roswell Park Memorial Institute containing 0.1% azide and 3% fetal calf serum and surface stained in 96-well plates with the 10–3.6 PE (anti-I-Ag7) (BD PharMingen, Franklin Lakes, NJ, USA), which is class II major histocompatibility complex haplotype for NOD/ShiLtJ mice. All samples were analyzed on a FACSCalibur flow cytometer (Becton Dickinson, Mountain View, CA, USA) using CellQuest software (Becton Dickinson).

### 2.4. Staining and Immunohistochemistry (IHC)

On serial heart sections and pancreatic sections obtained from F15 NOD *Nfκb1* wild-type mice, F15 NOD *Nfκb1* heterozygote mice, and F15 NOD *Nfκb1* homozygote mice, IHC staining for Cluster of Differentiation 3 (CD3) was performed (Appendix A) using mouse CD3 monoclonal antibody (catalog no.17A2, 1:200) purchased from Thermo Fisher Scientific (Waltham, MA, USA). IHC was performed using the avidin–biotin complex method, as previously described. Hematoxylin and eosin (H&E) staining was performed through standard procedure.

### 2.5. Vaccine Immunization

The influenza vaccine (20 or 50 μL, influenza HA vaccine “KMB”; KM Biologics Co. Ltd., Kumamoto-city, Kumamoto, Japan), HBV vaccine (Bimmugen; KM Biologics Co. Ltd., Kumamoto-city, Kumamoto, Japan), CERVARIX (GlaxoSmithKline plc, Borough of Hounslow, London, UK), GARDASIL (MSD K.K. Chiyoda, Tokyo, Japan), or phosphate-buffered saline (PBS) as immunogen was injected intramuscularly into the quadriceps femoris muscle of 10-week-old 129P2(B6)-*Nfκb1* wild-type, heterozygote, and homozygote mice, and F15 NOD *Nfκb1* wild-type, heterozygote, homozygote mice, as well as NOD/ShiLtj mice, for immunological studies, including cardiac studies and allergy assay. Mice were given the second doses of influenza HA vaccine, Bimmugen, CERVARIX, GARDASIL, or PBS, 30 days following the first shot of the influenza HA vaccine, Bimmugen, CERVARIX, GARDASIL, or PBS.

### 2.6. Cardiac Troponin T (cTnT) Assay and Myocarditis Scoring

To correlate serum cTnT levels with the incidence and severity of myocarditis, mice were anesthetized with ether and bled retro-orbitally, or blood was exclusively collected at autopsy. Serum cTnT levels were measured using a mouse cTnT enzyme-linked immunosorbent assay (ELISA) kit (Cusabio Technology LLC, Houston, TX, USA) according to the manufacturer’s instructions.

### 2.7. Multiplex Cytokine Bead Array Assay and Mouse IgG Assay

The cytokine and chemokine levels in mouse serum (20× dilution) were measured using the Mouse Cytokine Array Panel A (catalog no. ARY006; R&D Systems, Inc., Minneapolis, MN, USA) based on the manufacturer’s instructions. Sera were collected from each mouse at the time of dissection (Appendix A).

### 2.8. Naïve CD8^+^ T-Cell Isolation and Proliferation Assay for CD8^+^ T-Cells

Splenocytes from the mice were harvested and red blood cells were removed. The CD8^+^ T-cells were then sorted using a magnetic-activated cell sorting kit (CD8^+^ T-cell isolation kit II, Miltenyi Biotech, Bergisch Gladbach, Germany). CD8^+^ T-cells (1 × 10^6^ cells/mL) were cocultured with 100 μL of PBS containing a range of cardiac myosin heavy chain-α (amino acids 334–352 peptide, Cosmo Bio Co., Ltd., Koto-Ku, Tokyo, Japan) or Influenza HA (46–54) peptide (FMYSDFHFI) (M&S TechnoSystems, Inc., Osaka-city, Osaka, Japan) at a concentration of 5 or 10 μg/mL, mouse interleukin 2 (IL-2) at a concentration of 10 or 50 IU/mL, or mouse IL-12 at a concentration of 10 or 20 ng/mL (PeproTech, Shanghai, China) for 48 h. The cultured cells were harvested and the CD8^+^ T-cell proliferation was assessed using the BrdU cell proliferation ELISA kit (Abcam, Cambridge, MA, USA). The supernatant was collected and analyzed for interferon-gamma (IFN-γ) release.

### 2.9. Allergy Assay

The immunoglobulin E (IgE) levels in mouse serum (10× dilution) were measured using OpIEATM set mouse IgE (BD PharMingen, Lakes, NJ, USA) according to the manufacturer’s instructions. As an experimental allergy model, 10-week-old BALB/c mice (Japan Clear, Meguro-ku, Tokyo, Japan) were injected intraperitoneally with 100 μL of a solution containing formalin-treated *Pseudomonas pertucinogena* suspension (80 μg/body) and Mite Extract DF (50 μg/body).

### 2.10. Statistical Analysis

All data are expressed as the mean and standard error of the mean (SEM). Normality was verified using the Shapiro–Wilk test. For comparing two groups, the unpaired two-tailed *t* test or Mann–Whitney *U* test was used. Multiple comparisons were performed using a one-way analysis of variance with a Tukey post hoc test or a Kruskal–Wallis analysis with a post hoc Steel–Dwass or Steel test. A *p*-value of less than 0.05 was considered statistically significant. All statistical analyses were conducted using the JMP software (SAS Institute, Cary, NC, USA).

### 2.11. Ethical Approval and Consent to Participate

This study was reviewed and approved by the Central Ethics Review Board of the National Hospital Organization Headquarters in Japan (Tokyo, Japan) and Shinshu University (Nagano, Japan) on 17 August 2019, with approval codes NHO H31-02 and M192.

Details of Materials and Methods are indicated in Appendix A Sets available online.

## 3. Results

To understand the risks and consequences of myocarditis following vaccinations with many vaccines, including COVID-19, we examined an experimental system using genetically modified small animals. Because of the polygenetic nature of autoimmune diseases, autoimmune disease is difficult to spontaneously induce in genetically modified mice. Therefore, we investigated changes in the symptoms of an autoimmune disease based on the expression status of NF-κB1 in NOD/ShiLtJ mice, which was established as a mouse model for type 1 diabetes [12].

The F15-NOD *Nfκb1* heterozygote was created by backcrossing D2.129P2 (B6)-*Nfκb1* homozygote mice with NOD/ShiLtJ mice 15 times (Appendix A). The fact that F15-NOD *Nfκb1* heterozygote mice have a genetic background in NOD mice was confirmed by fluorescence-activated cell sorting (FACS) results of haplotype g7 expression, which is a genetic marker in NOD mice (Appendix A). Additionally, F15-NOD *Nfκb1* heterozygote male and female mice were crossed to create the F15 NOD *Nfκb1* homozygote. Accordingly, approximately 30% of all infant mice born from the identical F15 NOD *Nfκb1* heterozygote mother mice died by 9 weeks of age. Therefore, all infant mice were genotyped. Consequently, all of the dead infant mice were F15 NOD *Nfκb1* homozygotes (Figure 1A). Similar to NOD/ShiL.tj mice, 90% of female mice and 52% of male mice acquired diabetes at 30 weeks of age in the F15 NOD *Nfκb1* wild-type mice (Figure 1A). Conversely, all F15-NOD *Nfκb1* heterozygote mice, regardless of gender, developed insulitis by 10 months of age (Figure 1A). Histopathological examination and M-mode echocardiography revealed that both male and female infant mice died from myocarditis by 9 weeks of age (Figure 1B and Appendix A). In NOD/ShiLtJ mice, the incidence of insulitis in females is higher than that in males. However, insulitis was observed in all F15 NOD *Nfκb1* heterozygote mice (Appendix A). Furthermore, the *Nfκb1* expression was found to be involved in the onset of insulitis.

Based on these findings, F15-NOD *Nfκb1* heterozygote and homozygote mice are at risk of developing myocarditis. Therefore, we conducted histopathological studies in NOD/ShiLtj, *Nfκb1* genetically modified 129P2 (B6), and F15-NOD mice to investigate myocarditis development after inoculation with influenza vaccine (influenza HA vaccine “KMB”) or HBV vaccine (Bimmugen). In the F15-NOD *Nfκb1* wild-type mice, the infiltration of CD3-positive T-cells was not detected in myocardial tissue after inoculation with influenza or HBV vaccines (Figure 2A), but it was slight in F15-NOD *Nfκb1* heterozygote mice (Figure 2A). The average percentage of CD3-positive T-cells infiltrating myocardial tissues in the unvaccinated, influenza-vaccinated, and HBV-vaccinated groups was 6.14%, 23.92%, and 32.14%, respectively (Figure 2A). Myocarditis symptoms may be slightly worse after HBV vaccination compared with influenza vaccination. Additionally, we investigated cTnT, which is a marker for myocarditis, after the inoculation of influenza or HBV vaccine in NOD/ShiLtj, *Nfκb1* genetically modified 129P2 (B6), and F15-NOD mice. Consequently, an increase in cTnT level (>0.014 ng/mL) due to influenza or HBV vaccination was observed with no gender difference (Figure 2B). In 129P2 (B6) wild-type and 129P2 (B6) *Nfκb1* heterozygote mice, the cTnT level after influenza or HBV vaccination was below the standard (0.014 ng/mL) with no gender difference (Figure 2B). In NOD/ShiLtj and F15 NOD *Nfκb1* wild-type mice, the median cTnT level for PBS, influenza vaccine, or HBV vaccine ranged from 0.012 to 0.014 ng/mL. In F15 NOD *Nfκb1* heterozygote mice, an increase in cTnT level (>0.02 ng/mL) after influenza or HBV vaccination was detected with no gender difference (Figure 2B). Based on the pathological diagnosis tests and serum cTnT results, a small proportion of infant mice with F15 NOD *Nfκb1* heterozygote are suspected to have mild myocarditis but mature normally like the F15 NOD *Nfκb1* wild-type mice (Figure 1 and Figure 2). The F15 NOD *Nfκb1* heterozygote and homozygote mice were found to have an increased risk of myocarditis after influenza or HBV vaccination.

There was no difference in serum cTnT levels between the 20 and 50 μL vaccinated F15 NOD *Nfκb1* heterozygote mice (Appendix A). However, in F15 NOD *Nfκb1* heterozygote mice inoculated with PBS as a control, an increase in IFN-γ, which is a type 1 immune response, was observed (Appendix A). In other words, the 20 μL vaccine, like the 50 μL vaccine, is believed to activate host immunity as an immunogen without triggering an autoimmune response.

In clinical practice, anaphylactic shock and encephalomyelitis have been reported as side effects of HPV vaccines, such as CERVARIX or GARDASIL. However, the development of cardiovascular disorders, such as myocarditis and endocarditis, as a side effect of CERVARIX or GARDASIL vaccination is extremely rare. To identify the incidence of myocarditis, 50 μL of CERVARIX or GARDASIL was inoculated into the quadriceps femoris muscle of male and female mice with F15 NOD *Nfκb1* heterozygotes, and histopathological studies were conducted and serum cTnT levels were measured (Appendix A). Our findings demonstrated no evidence of myocarditis development after CERVARIX or GARDASIL inoculation (Appendix A). This result implies that side effects experienced in clinical practice are typical.

A recent study demonstrated that IFN-γ and tumor necrosis factor-alpha (TNF-α) levels increased steadily in infected D2.129P2 (B6)-*Nfκb1* wild-type mice, peaking at day 10 post-infection (p.i.) and declining to uninfected levels by day 20 p.i. [13]. Similarly, in D2.129P2 (B6)-*Nfκb1* homozygote mice, IFN-γ and TNF-α levels increased [13]. However, on day 20 p.i., the levels of these cytokines were significantly higher in D2.129P2 (B6)-*Nfκb1* homozygote mice than in D2.129P2 (B6)-*Nfκb1* wild-type mice (*p* < 0.05), indicating a lack of clearance of viral infection, such as *C. rodentium* infection [13].

Thus, the inflammatory cytokine levels in serum were measured using a Multiplex cytokine bead array assay and they were found to be increased in mice after influenza or HBV vaccination (Figure 3 and Appendix A). After each vaccine, the serum levels of inflammatory cytokines were increased in F15-NOD *Nfκb1* heterozygote mice compared with the F15-NOD *Nfκb1* wild-type mice (Appendix A). It was revealed that serum IFN-γ levels were significantly increased, especially in mice after influenza or HBV vaccination (Figure 3 and Appendix A). Compared with the influenza vaccination, HBV vaccination strongly induces the expression of inflammatory cytokines in mouse serum (Figure 3 and Appendix A). Additionally, it was revealed that serum IgG levels were significantly increased in mice after influenza or HBV vaccination. Furthermore, vaccination-induced serum IgG levels in F15-NOD *Nfκb1* heterozygote mice were slightly lower than those in F15 NOD *Nfκb1* wild-type mice (Appendix A), which is consistent with previous studies [5,10].

The cytokine assay results demonstrated that plasma IFN-γ levels were markedly increased, particularly in mice after influenza or HBV vaccination (Figure 3). IFN-γ is secreted by the antigen-recognized activated CD8-positive T-cells, such as cytotoxic T-cells (CTLs). Therefore, CTLs were subpurified from various mice inoculated with influenza or HBV vaccine using a magnetic-activated cell sorting kit. Subpurified CTLs were cocultured for 48 h with the mouse myosin heavy chain peptide identified as an epitope. After 48 h of coculture, the CD8-positive T-cell proliferation was assessed using the BrdU cell proliferation ELISA kit. Additionally, the supernatant was harvested and analyzed for IFN-γ release using ELISA.

Subpurified CTLs from mice were cocultured for 48 h with the mouse myosin heavy chain peptide identified as an epitope. After 48 h of coculture, CTLs subpurified from mice vaccinated with influenza or HBV vaccine were found to be more strongly activated and proliferated when cocultured with myosin heavy chain peptide than those subpurified from mice inoculated with PBS as a negative control (Figure 4). Additionally, CTLs subpurified from mice vaccinated with HBV vaccine were more strongly activated and proliferated when cocultured with myosin heavy chain peptide than those subpurified from mice vaccinated with influenza vaccine (Figure 4). The CTLs subpurified from HBV-vaccinated male mice of the F15-NOD *Nfκb1* heterozygote are the most strongly activated and proliferated when cocultured with myosin heavy chain peptide (Figure 4). Coculture with myosin heavy chain peptide activates CTLs, as evidenced by increased expression and secretion of IFN-γ (Figure 4). The results of these immune system experiments indicate that inoculating F15-NOD *Nfκb1* heterozygote mice with influenza or HBV vaccine may cause myocarditis. CTLs subpurified from PBS-inoculated F15-NOD *Nfκb1* heterozygote mice have been shown to be activated and proliferated via IL-2 stimulation (Figure 4), indicating that the BrdU cell proliferation ELISA kit is working properly.

Subpurified CTLs were cocultured for 48 h with the influenza virus HA peptide identified as an epitope. After 48 h of coculture, CTLs subpurified from mice vaccinated with influenza vaccine were found to be significantly activated and proliferated when cocultured with the influenza virus HA peptide compared with those subpurified from mice inoculated with PBS or HBV vaccine (Appendix A). Additionally, CTLs subpurified from PBS-inoculated F15-NOD *Nfκb1* heterozygote mice have been shown to be activated and proliferated via IL-12 stimulation, which reportedly enhanced myocarditis [14], indicating that the BrdU cell proliferation ELISA kit is working properly (Appendix A).

In many cases, myocarditis was observed to develop within 5 days after vaccination. The short time course suggests an allergic mechanism, probably because of the formation of immune complexes. Therefore, we investigated the possibility that influenza or HBV vaccination induces allergic symptoms in various genetically modified mice. Serum was collected from each mouse vaccinated with influenza vaccine, HBV vaccine, or PBS, and serum IgE levels were measured using the OpIEATM set mouse IgE kit. Based on IgE results, there was no evidence that influenza or HBV vaccination could induce allergic symptoms in various genetically modified mice (Appendix A). However, the serum IgE level was found to be significantly increased in experimental allergy-induced mice (Appendix A), indicating that the OpIEATM set mouse IgE kit is working properly.

## 4. Discussion

NOD/ShiLtJ mice, like biobreeding rats, are used as an animal model for type 1 diabetes. Diabetes develops in NOD mice as a result of insulitis, a leukocytic infiltrate of the pancreatic islets. The onset of diabetes is associated with moderate glycosuria and nonfasting hyperglycemia. Previously, in NOD/ShiLtJ mice that spontaneously developed type 1 diabetes, the possible involvement of decreased expression of LMP2/β1i, an immunoproteasome β subunit, and associated decreased expression of NF-κB1 (also known as p50) in the development of type 1 diabetes was investigated. In response to these arguments, NOD mice with inconsistent NF-κB1 expression were established. Surprisingly, the majority of NOD *Nfκb1* homozygote mice were found to die by the eighth week of life because of severe myocarditis. The incidence of spontaneous myocarditis in mice was slightly higher in males than in females. Furthermore, insulitis was observed in all NOD *Nfκb1* heterozygote mice as early as 4 months of age. Additionally, in NOD *Nfκb1* heterozygote mice, myocarditis with an increase in cTnT levels due to influenza or HBV vaccination was observed with no significant gender difference. However, myocarditis was not detected with the two types of HPV vaccination. The results of immunological assays and histopathological examinations indicated that vaccination could induce myocarditis in genetically modified mice. Furthermore, we discovered that decreased NF-κB1 expression is directly involved in the development of autoimmune diseases, such as myocarditis or insulitis in NOD/ShiLtJ mice (Appendix A).

Our findings indicate that defective NF-κB1 expression is involved in the development of myocarditis, an autoimmune disease. Previous clinical studies have linked the development of ulcerative colitis to reduced NF-κB1 expression because of pathological variants. Generally, immunosuppressants and/or anti-TNF-α antibodies are commonly used to treat patients with ulcerative colitis [15,16,17]. Therefore, it has been established that the antivirus antibody production in response to vaccines, including the COVID-19 mRNA vaccine, is lower in patients with ulcerative colitis than in healthy subjects. In other words, even if NF-κB1 expression is reduced in patients with ulcerative colitis, it is unlikely that patients with ulcerative colitis will develop myocarditis after COVID-19 mRNA vaccination while being treated with immunosuppressive drugs. Myocarditis may develop after COVID-19 mRNA vaccination in individuals who do not develop autoimmune disease and may have reduced NF-κB1 expression.

According to a recent study, NF-κB1 negatively regulates TNF-α production in resting dendric cells and prevents the subsequent induction of CD8-positive effector activity [18]. Previous research has reported that the expressions of inflammatory cytokines, such as TNF-α and CD8-positive T-cell proliferation, were significantly increased in virus-infected *Nfκb1*-heterozygote mice compared with *Nfκb1* wild-type mice. The immunological results revealed that the inflammatory cytokine expression and CD8-positive T-cell proliferation were increased in the vaccinated *Nfκb1*-heterozygote mice compared with *Nfκb1* wild-type mice. Presumably, viral structural proteins and/or adjuvants in various vaccines might activate the type I immune response. Furthermore, because of low *Nfκb1* expression, the components of these vaccines are believed to significantly activate the type I immune response. Although the mechanism of immune response to these antigens is unknown, *Nfκb1* has a structure homologous to the I-κB family, which is a suppressor of immune activity. Furthermore, immunity may also be activated by *Nfκb1*deficiency.

Our findings demonstrate that HBV vaccination (Bimmugen) is more likely to cause myocarditis in genetically modified mice than influenza HA vaccination (KMB). Aluminum and/or CpG, which are components of vaccines, activate antigen-presenting cells, such as macrophages and innate immunity. Because many people have been infected with the influenza virus, they have already developed immunity to it. Therefore, influenza HA vaccines (KMB) manufactured by Japanese pharmaceutical companies and used in clinical practice do not contain adjuvants, such as aluminum or CpG. However, the HBV vaccine (Bimmugen), manufactured by a Japanese pharmaceutical company, contains aluminum as an adjuvant. Therefore, the HBV vaccine (Bimmugen) is more likely to activate the host immune response than the influenza HA vaccine (KMB). However, although HPV vaccines, such as CERVARIX and GARDASIL, contain aluminum as an adjuvant, the incidence of cardiovascular side effects, such as myocarditis, may be extremely low (Appendix A). Therefore, components other than adjuvant in the vaccine may induce the development of cardiovascular disorders.

There was no significant difference in serum cTnT levels observed between the 20 and 50 μL vaccinated F15 NOD *Nfκb1* heterozygote mice. However, in F15 NOD *Nfκb1* heterozygote mice inoculated with PBS as a control, an increase in IFN-α, which is a type 1 immune response, was observed. In other words, the 20 μL vaccine, like the 50 μL vaccine, is believed to activate host immunity as an immunogen without triggering an autoimmune response. Therefore, the effects of low-concentration vaccines should be studied in new clinical trials so that vaccines can be safely given to people who are predisposed to autoimmune diseases or allergies.

Five decades ago, a man in his twenties developed myocarditis after receiving a smallpox vaccine made from the attenuated vaccinia virus [19,20,21,22,23]. Since then, myocarditis has been sporadically reported following various vaccinations [24]. Since myocarditis has been observed following vaccination with an attenuated virus or other main components, the etiology of myocarditis observed after mRNA-based COVID-19 vaccination is not the genome SARS-CoV-2 mRNA. The described myocarditis cases typically occurred 10–14 days after a primary smallpox vaccination. Furthermore, in the COVID-19 era, myocarditis occurred in many cases within 5 days of mRNA-based COVID-19 vaccination [25]. This time course suggests an allergic mechanism, probably because of the formation of immune complexes. An autoimmune mechanism has also been proposed for cardiac-related adverse reactions following HPV vaccination [26,27,28]. Unfortunately, the role of immunosuppressive therapy remains unclear.

The prevalence and mortality of cardiovascular diseases, such as ischemic heart disease, myocarditis, and endocarditis, are found to be higher in males than in females. The incidence of cardiovascular disease is 4.1/1000 people per year for females and 6.4/1000 people per year for males, which is 25% lower in females [27]. Additionally, females have a 38% lower mortality rate and a 27% lower recurrence risk of cardiovascular disorders than males [27,28]. Various protective effects on cardiovascular disorders, such as female hormones, are considered a cause of gender differences in the incidence and mortality of such cardiovascular disorders.

Our findings revealed that the risk of spontaneous myocarditis was slightly higher in male F15 NOD *Nfκb1* homozygote mice (95%, 19/20) than in female F15 NOD *Nfκb1* homozygote mice (75%, 15/20) (Appendix A). These results support the clinical practice finding that the prevalence of myocarditis after influenza HA or HBV (Bimmugen) vaccination is slightly higher in men than in women, as evidenced in clinical practice. Chest pain and myocarditis were noted in 5166 (BNT162b2) and 399 (mRNA-1273) recipients in a clinical trial conducted on more than 129 million people vaccinated with the BNT162b2 or mRNA-1273 vaccine [29]. In clinical trials that examined the incidence of chest pain and myocarditis by age, the highest rates observed were in young male individuals (ages 12–17 years) after the second doses of vaccination [30]. The results of our study also showed that the incidence of myocarditis in male mice after vaccination may be higher than that in females. The reason for the gender difference in the incidence of myocarditis observed after inoculation with mRNA-based vaccines has not been clarified. Additionally, female hormones may play a role in the gender differences in the onset of cardiovascular disease after vaccination. However, the mechanism underlying the development of cardiovascular disorders following vaccination remains unclear. In clinical practice, gender differences in the development of autoimmune diseases, such as rheumatoid arthritis, have been reported. Further research in these genetically modified mice may reveal the pathogenic mechanism underlying myocarditis development as well as gender differences following mRNA-based COVID-19 vaccination.

In this study, we used genetically modified mice, but not all of the results acquired from genetically modified mice are reflected in human symptoms. Therefore, these research results should be verified using human samples. Treatment for vaccination-induced myocarditis is given for each patient’s symptoms, but the efficacy of immunosuppressive therapy is unclear. Clinical studies conducted worldwide have shown a higher incidence of myocarditis in young men following second doses of the COVID-19 mRNA vaccine than in women [31,32,33,34]. Importantly, the COVID-19 vaccine has a very favorable risk-to-benefit ratio for all age and gender groups evaluated so far. However, because of the limited data on the efficacy and safety of mRNA-based COVID-19 vaccines in patients with autoimmune diseases, such as collagen and rheumatic diseases, mRNA-based COVID-19 vaccination for patients with autoimmune diseases or people with allergic predisposition should be carefully considered.

## 5. Conclusions

Using autoimmune-onset model mice, our study revealed that NF-κB1 is involved in the development of myocarditis. Additionally, spontaneous myocarditis was observed in the genetically modified mice following vaccination. Our findings will provide new insights into the risk factors involved in the development of cardiovascular disorders, such as myocarditis, after vaccination, as well as the development of clinical treatments for these disorders.

## Figures and Tables

**Figure 1 biomedicines-10-01443-f001:**
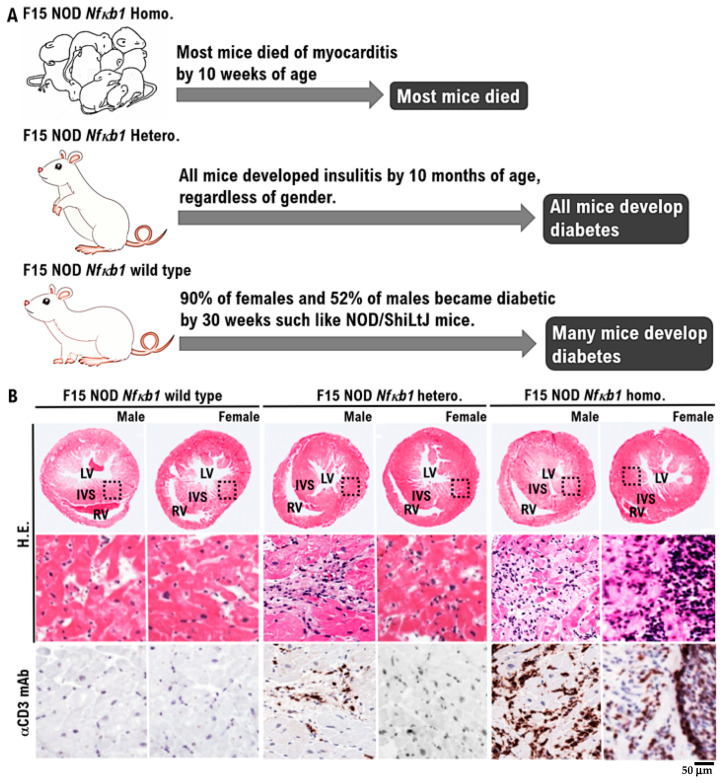
The onset of spontaneous myocarditis in F15 NOD *Nfκb1* homozygote mice. (**A**). The phenotype of F15 NOD mice is determined based on the expression status of the *Nfkb1* gene. By 10 weeks of age, the majority of F15 NOD *Nfκb1* homozygote mice had died because of myocarditis. The incidence of spontaneous myocarditis was found to be slightly higher in male F15 NOD *Nfκb1* homozygote mice (incident rate: 95%, 19/20) than in female F15 NOD *Nfκb1* homozygote mice (incident rate: 75%, 15/20). By 10 months of age, all F15 NOD *Nfκb1* heterozygote mice developed insulitis, regardless of gender. In the case of F15 NOD *Nfκb1* wild-type mice, 90% of females and 52% of males developed diabetes by 30 weeks, similar to NOD/ShiLtJ mice. (**B**). Spontaneous myocarditis in F15 NOD *Nfκb1* heterozygote and homozygote male and female infant mice by 9 weeks of age. F15 hybrids of *Nfκb1* heterozygote (hetero.), homozygote (homo.), and wild-type mice with a nonobese diabetic/ShiLtJ background are shown. Histopathological examination revealed that both male and female infant mice died by 9 weeks of age because of myocarditis. Representative hematoxylin and eosin (H&E)-stained tissue sections and immunohistochemically stained tissue sections with anti-CD3 monoclonal antibodies of the hearts obtained from both male and female infant mice by age 4–8 weeks are shown. The immunohistology images show diffuse infiltration of CD3 positive T-cells, as indicated by anti-CD3 antibody staining (brown) in the lower panels. All subtypes of T cells always express the CD3 antigen on their surface. Images in upper panels ×40. Images in lower panels ×400. Experiments were conducted with 20 animals in each group. RV; right ventricle, LV; left ventricle, IVS; interventricular septum.

**Figure 2 biomedicines-10-01443-f002:**
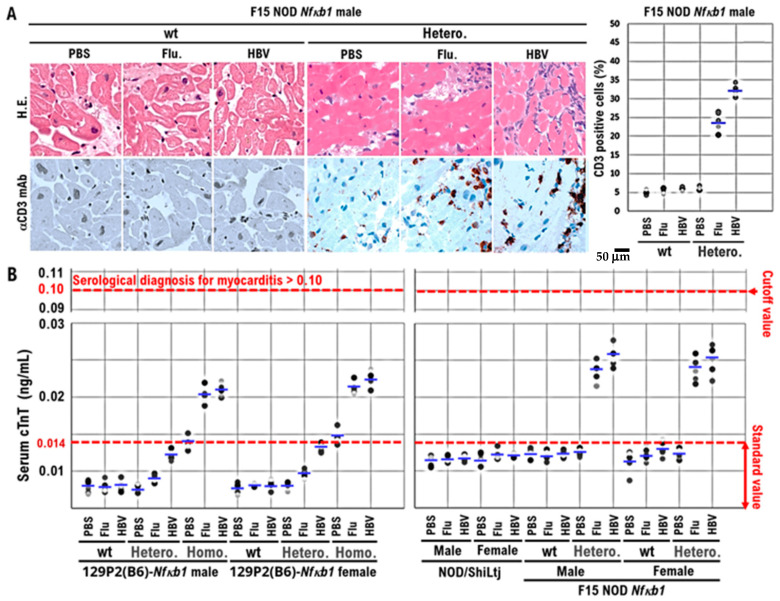
The onset of myocarditis in F15 NOD *Nfκb1* heterozygote mice after vaccination with influenza HA or HBV vaccine (Bimmugen). (**A**) Representative hematoxylin and eosin (H&E)-stained tissue sections and immunohistochemically stained tissue sections with anti-CD3 monoclonal antibodies of the hearts obtained from both F15 NOD *Nfκb1* wild-type male mice and F15 NOD *Nfκb1* heterozygote male mice. The myocarditis was not observed in F15 NOD *Nfκb1* heterozygote and wild-type male mice after intramuscular injection with 50 μL of influenza vaccine (influenza HA vaccine “KMB”), 50 μL of HBV vaccine (Bimmugen), or 50 μL of PBS. After 30 days after the first dose immunization of the influenza HA vaccine, Bimmugen, or PBS, the mice received second doses of the influenza HA vaccine, Bimmugen, or PBS. The blood and heart tissues were collected at the time of dissection (Appendix A). Images ×400. (**B**) Increased serum cardiac troponin T (cTnT) level in 129P2 (B6)-*Nfκb1* homozygote mice and F15-NOD *NfκB1* heterozygote mice after influenza HA or HBV vaccination. PBS (as a control), influenza HA vaccine, or HBV vaccine were inoculated into the quadriceps femoris muscle of the left thigh muscle of 10-week-old 129P2 (B6)-*Nfκb1* wild-type, 129P2(B6)-*Nfκb1* heterozygote, 129P2(B6)-*Nfκb1* homozygote, NOD/ShiLtj, F15-NOD *Nfκb1* wild-type, and F15-NOD *Nfκb1* heterozygote mice. Influenza HA or HBV vaccine (Bimmugen) was given to mice (100 μL) after diluting the stock solution fivefold with PBS. In 129P2 (B6)-*Nfκb1* homozygote and F15-NOD *Nfκb1* heterozygote mice inoculated with influenza or HBV vaccine, the serum cTnT level was ≥0.02 ng/mL, with no gender difference. Serum cTnT levels were measured using the mouse cTnT ELISA Kit (Cusabio Technology LLC) according to the manufacturer’s instructions. The standard serum cTnT concentration is ≤0.014 ng/mL. In clinical practice, myocarditis is diagnosed when the serum cTnT concentration is ≥0.1 ng/mL. Each mouse group consisted of five mice. Experiments were conducted with 5 animals in each group.

**Figure 3 biomedicines-10-01443-f003:**
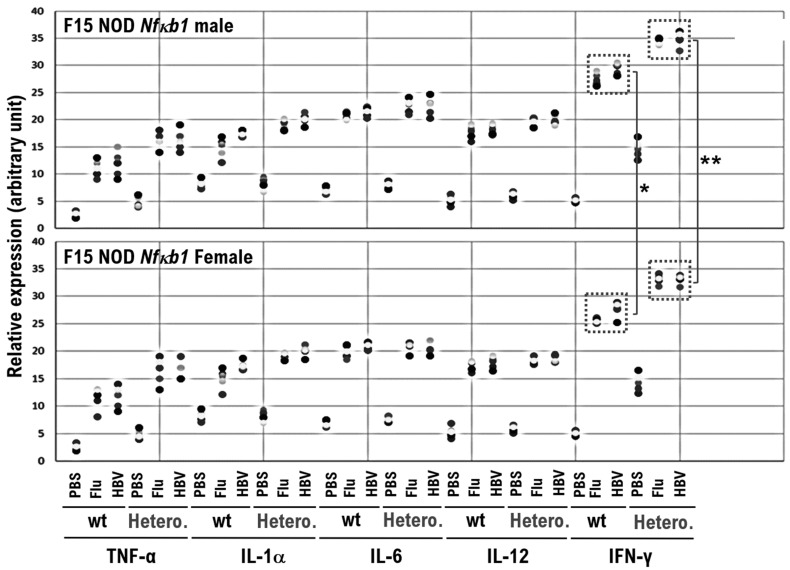
Elevated serum inflammatory cytokine levels in F15 NOD *Nfκb1* wild-type mice and F15 NOD *Nfκb1* heterozygote mice after vaccination with influenza HA or HBV (Bimmugen) vaccine. To collect serum for measuring serum inflammatory cytokine levels, mice were anesthetized with ether and bled retro-orbitally or alternatively 3 days after inoculation with 50 μL of influenza HA or HBV (Bimmugen) vaccine, or PBS. The cytokine and chemokine levels in mouse serum (20× dilution) were measured using the Mouse Cytokine Array Panel A (catalog no. ARY006; R&D Systems, Inc.) based on the manufacturer’s instructions. The serum inflammatory cytokine levels in male mice (upper panel) and female mice (lower panel) are shown. The serum inflammatory cytokine levels in F15 NOD *Nfκb1* wild-type and F15 NOD *Nfκb1* heterozygote mice after vaccination were significantly elevated 3 days after vaccination, but no elevation was observed in F15 NOD *Nfκb1* wild-type and F15 NOD *Nfκb1* heterozygote mice after intramuscular injection of PBS as a negative control. The levels of IFN-γ, an inflammatory cytokine that is markedly released from activated CD8-positive T-cells (also known as CTLs), were higher in male mice than in female mice. TNF-α, IL-1α, IL-6, and IFN-γ are inflammatory cytokines that are regulated by the NF-κB signaling cascade. The levels of all listed inflammatory cytokines are shown in Appendix A. * *p* < 0.01, ** *p* < 0.005. Experiments were conducted with 5 animals in each group.

**Figure 4 biomedicines-10-01443-f004:**
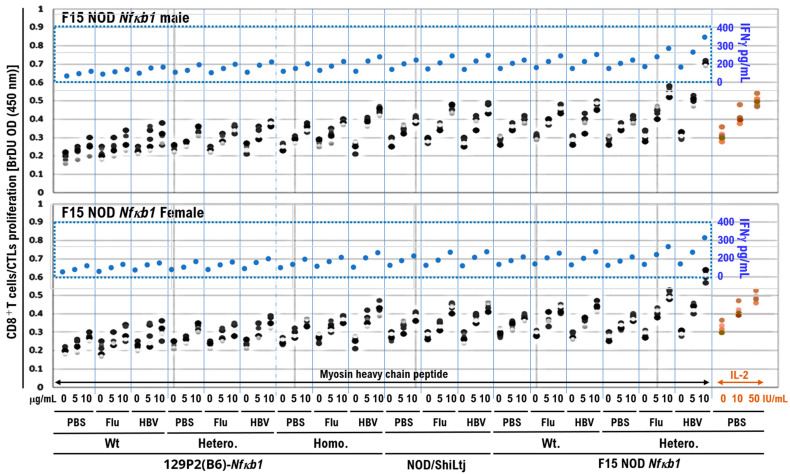
Activation of subpurified CD8^+^ T-cell proliferation in coculture with Myosin heavy chain peptide. Splenocytes from the mice immunized by intramuscular injection of 50 μL of influenza HA vaccine, 50 μL of HBV vaccine (Bimmugen), or PBS were harvested, and red blood cells were removed. CD8^+^ T-cells were then sorted using a magnetic-activated cell sorting kit (CD8^+^ T-cell isolation kit II, Miltenyi Biotech) according to the manufacturer’s protocol. The isolated CD8^+^ T-cells were resuspended in PBS containing 2% FBS. CD8^+^ T-cells (1 × 10^6^ cells/mL) were cocultured with 100 μL of PBS containing a range of cardiac myosin heavy chain-α amino acids 334–352 peptide (Cosmo Bio Co., Ltd.) at a concentration of 5 or 10 μg/mL, mouse IL-2 at a concentration of 10 or 50 IU/mL (PeproTech) for 48 h. The cultured cells were harvested and the CD8^+^ T-cell proliferation was assessed using the BrdU cell proliferation enzyme-linked immunosorbent assays (ELISA) kit (catalog no. ab126556; Abcam, Cambridge) based on the manufacturer’s instructions. The supernatant was collected and analyzed for IFN-γ release. IFN-γ levels in the culture supernatants were measured via ELISA using a commercial kit (R&D Systems, Inc.) based on the manufacturer’s instructions. Coculture of cardiac myosin heavy chain-α aa334–352 peptide with CD8^+^ T-cells subpurified from the F15 NOD *Nfκb1* heterozygote mice inoculated with influenza HA or HBV vaccine markedly activated the CD8^+^ T-cell proliferation and increased serum IFN-γ levels. Experiments were conducted with 5 animals in each group.

## Data Availability

The study did not report any data.

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
