# Peer review of "Spontaneous Myocarditis in Mice Predisposed to Autoimmune Disease: Including Vaccination-Induced Onset"

_biomedicines, 2022, doi:10.3390/biomedicines10061443_

Round 1
Reviewer 1 Report
Hayashi and colleagues describe a mouse model of spontaneous myocarditis. The mice are predisposed to autoimmune disease and myocarditis can be provoked by different models of vaccination. The manuscript is well written, and the data clearly and very carefully presented.
I have only some small criticisms:
The kappa in NFkappaB is missing throughout the text.
Chapter 2.9: “(80 g/body)” Symbol is missing. This problem continues throughout the entire text.
Figure 1B: Please select for all images the same region – either septum or left ventricular free wall.
Author Response
Responses to each reviewer's comments
Manuscript ID: biomedicines-1780177
Reviewer 1:
Comments and Suggestions for Authors
Hayashi and colleagues describe a mouse model of spontaneous myocarditis. The mice are predisposed to autoimmune disease and myocarditis can be provoked by different models of vaccination. The manuscript is well written, and the data clearly and very carefully presented.
I have only some small criticisms:
Comment 1: The kappa in NFkappaB is missing throughout the text.
Answer 1: We appreciate your comment. We agree with your comment. We have revised the entire manuscript according to your comments. Since the kappa (k) is missing, we rewrote it to NF-kB1 or Nfkb1.
Comment 2: Chapter 2.9: “(80 g/body)” Symbol is missing. This problem continues throughout the entire text.
Answer 2: We appreciate your comment. We agree with your comment. We have revised the entire manuscript according to your comments. Since the micro is missing, we rewrote it to μg or μL.
Comment 3: Figure 1B: Please select for all images the same region – either septum or left ventricular free wall.
Answer 3: We appreciate your comment. We agree with your comment. We have revised the Figure 1B according to your comments. We also added additional matters (RV; Right ventricle, LV; Left ventricle, IVS; Interventricular septum) to legend section of Figure 1.
RV; Right ventricle, LV; Left ventricle, IVS; Interventricular septum.
We also added additional material information regarding sections of heart tissues in Supplementary materials as followings.
The section used for Immunohistochemistry staining is a slice obtained from a cross section cut in a horizontal section at the position of the mouse heart (4.5 mm from the top) shown in the figure below.

Reviewer 2 Report
Proposed paper is interesting and well written. However, I suggest two minor issue inorder to improve the paper:
- Please report the numbr of experimental animal in all the figure were applicable.
- Please add recently reference on COVID-19 vaccinatino vaccines to your reference and better discuss this issue un the introduction or in the discussion section. CFR: Circulation. 2021 Aug 10;144(6):506-508. and J Am Coll Cardiol. 2022 May 3;79(17):1717-1756.
Author Response
Responses to each reviewer's comments
Manuscript ID: biomedicines-1780177
Reviewer 2:
Comments and Suggestions for Authors
Proposed paper is interesting and well written. However, I suggest two minor issue in order to improve the paper:
Comment 1: Please report the number of experimental animals in all the figure were applicable.
Answer 1: We appreciate your comment. We agree with your comment. We have revised the entire manuscript according to your comments. Experiments were conducted with 20 animals in each group (Figure 1). Experiments were conducted with 5 animals in each group (Figure 2, Figure 3, Figure 4).
Comment 2: Please add recently reference on COVID-19 vaccinatino vaccines to your reference and better discuss this issue un the introduction or in the discussion section. CFR: Circulation. 2021 Aug 10;144(6):506-508. and J Am Coll Cardiol. 2022 May 3;79(17):1717-1756.
Answer 2: We appreciate your comment. We agree with your comment. We have revised the manuscript according to your comments. We reviewed the content of the two reports on the COVID-19 vaccine presented by the reviewer and added the issues to the discussion section as follows. We have added two reports on the COVID-19 vaccine presented by the reviewers to the references in the revised manuscript.
Chest pain and myocarditis were noted in 5166 (BNT162b2) and 399 (mRNA-1273) recipients in a clinical trial conducted on more than 129 million people vaccinated with the BNT162b2 or mRNA-1273 vaccine (29). In clinical trials that examined the incidence of chest pain and myocarditis by age, the highest rates observed were in young male individuals (ages 12-17 years) after the second doses of vaccination (30). The results of our study also showed that the incidence of myocarditis in male mice after vaccination may be higher than that in females. The reason for the gender difference in the incidence of myocarditis observed after inoculation with mRNA-based vaccine has not been clarified. Additionally, female hormones may play a role in the gender differences in the onset of cardiovascular disease after vaccination. However, the mechanism underlying the development of cardiovascular disorders following vaccination remains unclear. In clinical practice, gender differences in the development of autoimmune diseases such as rheumatoid arthritis have been reported. Further research in these genetically modified mice may reveal the pathogenic mechanism underlying myocarditis development as well as gender differences following mRNA-based COVID-19 vaccination.
Importantly, the COVID-19 vaccine has a very favorable risk-to-benefit ratio for all age and gender groups evaluated so far. However, because of the limited data on the efficacy and safety of mRNA-based COVID-19 vaccines in patients with autoimmune diseases such as collagen and rheumatic diseases, mRNA-based COVID-19 vaccination for patients with autoimmune diseases or people with allergic predisposition should be carefully considered.
- Larson, K.F.; Ammirati, E.; Adler, E.D.; Cooper, L.T.Jr.; Hong, K.N.; Saponara, G.; Couri, D.; Cereda, A.; Procopio, A.; Cavalotti, C.; Oliva, F.; Sanna, T.; Ciconte, V.A.; Onyango, G.; Holmes, D.R.; Borgeson, D.D. Myocarditis After BNT162b2 and mRNA-1273 Vaccination. Circulation. 2021 144(6), 506-508. doi: 10.1161/CIRCULATIONAHA.121.055913.
- Writing Committee.; Gluckman, T.J.; Bhave, N.M.; Allen, L.A.; Chung, E.H.; Spatz, E.S.; Ammirati, E.; Baggish, A.L.; Bozkurt, B.; Cornwell, W.K.3rd.; Harmon, K.G.; Kim, J.H.; Lala, A.; Levine, B.D.; Martinez, M.W.; Onuma, O.; Phelan, D.; Puntmann, V.O.; Rajpal, S.; Taub, P.R.; Verma, A.K. 2022 ACC Expert Consensus Decision Pathway on Cardiovascular Sequelae of COVID-19 in Adults: Myocarditis and Other Myocardial Involvement, Post-Acute Sequelae of SARS-CoV-2 Infection, and Return to Play: A Report of the American College of Cardiology Solution Set Oversight Committee. J Am Coll Cardiol. 2022 79(17), 1717-1756. doi: 10.1016/j.jacc.2022.02.003.
The BNT162b2 mRNA (Pfizer-BioNTech) and mRNA-1273 (Moderna) coronavirus disease 2019 (COVID-19) vaccines have gained widespread use across the globe to prevent further spread of severe acute respiratory syndrome coronavirus 2 (SARS-CoV-2) infection. Early studies and surveillance data suggest these vaccines are associated with no significant adverse events other than very rare anaphylaxis. Surveillance for other reactions continues. Myocarditis and inflammatory myocardial cellular infiltrate have been reported after vaccination, especially after the smallpox vaccine. However, myocarditis occurring after the BNT162b2 mRNA and mRNA-1273 vaccines has not been reported in trials. Here, we describe 8 patients who were hospitalized with chest pain and who were diagnosed with myocarditis by laboratory and cardiac magnetic resonance imaging within 2 to 4 days of receiving either the BNT162b2 or mRNA-1273 vaccine. In fact, the Centers for Disease Control’s Vaccine Adverse Event Reporting System (www.wonder.cdc.gov/vaers.html) received reports of chest pain and myocarditis in 5166 and 399 recipients, respectively, of the BNT162b2 or mRNA-1273 vaccine, whereas more than 129 million people have been fully vaccinated with these 2 vaccines. In conclusion, providers should be vigilant for myocarditis after COVID-19 mRNA vaccination, and further research is required to understand the long-term cardiovascular risks. Circulation. 2021 Aug 10;144(6):506-508.
Myocarditis following COVID-19 mRNA vaccination is an entity separate from but related to myocarditis following SARS-CoV-2 infection.119 In case reports and case series of vaccine-associated myocarditis, chest pain has been noted in the vast majority,12 occurring most commonly 2-3 days after the second mRNA vaccine dose. As of June 11, 2021, the reported rate of myocarditis following administration of the second dose of the COVID-19 mRNA vaccine was 40.6 cases per million in male individuals aged 12-29 years and 2.4 cases per million in male individuals aged >30 years. Most reported systemic reactions have been mild and transient, albeit more common among younger individuals and after the second vaccine dose. Importantly, a very favorable benefit-to-risk ratio exists with the COVID-19 vaccine for all age and sex groups evaluated thus far. Myocarditis following COVID-19 mRNA vaccination is rare. The highest observed rates have been in young male individuals (aged 12-17 years) after the second vaccine dose. Individuals presenting with chest pain early after receiving the COVID-19 mRNA vaccine should be evaluated for possible myocarditis. Initial testing should include an ECG, measurement of cTn, and an echocar diagram. J Am Coll Cardiol. 2022 May 3;79(17):1717-1756.
